# Advanced Photocatalytic Uranium Extraction Strategies: Progress, Challenges, and Prospects

**DOI:** 10.3390/nano13132005

**Published:** 2023-07-06

**Authors:** Wangchuan Zhu, Xiang Li, Danjun Wang, Feng Fu, Yucang Liang

**Affiliations:** 1Research Institute of Comprehensive Energy Industry Technology, School of Chemistry & Chemical Engineering, Yan’an University, Yan’an 716000, China; zhuwangchuan@163.com (W.Z.); yaulx418@163.com (X.L.); 2Institute of Inorganic Chemistry, University of Tübingen, Auf der Morgenstelle 18, 72076 Tübingen, Germany

**Keywords:** photocatalysis, structures, U(VI) extraction, mechanism, applications

## Abstract

Nuclear energy with low carbon emission and high-energy density is considered as one of the most promising future energy sources for human beings. However, the use of nuclear energy will inevitably lead to the discharge of nuclear waste and the consumption of uranium resources. Therefore, the development of simple, efficient, and economical uranium extraction methods is of great significance for the sustainable development of nuclear energy and the restoration of the ecological environment. Photocatalytic U(VI) extraction technology as a simple, highly efficient, and low-cost strategy, received increasing attention from researchers. In this review, the development background of photocatalytic U(VI) extraction and several photocatalytic U(VI) reduction mechanisms are briefly described and the identification methods of uranium species after photocatalytic reduction are addressed. Subsequently, the modification strategies of several catalysts used for U(VI) extraction are summarized and the advantages and disadvantages of photocatalytic U(VI) extraction are compared. Additionally, the research progress of photocatalytic technology for U(VI) extraction in actual uranium-containing wastewater and seawater are evaluated. Finally, the current challenges and the developments of photocatalytic U(VI) extraction technology in the future are prospected.

## 1. Introduction

In the context of global climate change, accelerating the transformation of clean and low-carbon energy became a global development trend [1]. As a mature technology, nuclear fission can produce stable output and the lowest carbon emissions, making a great contribution to meet the demand for low-carbon energy in the 21st century [2]. With the introduction of nuclear energy, the demand for uranium resources will inevitably increase. In a properly operated nuclear power plant, there should not be a continuous release of radioactive wastewater, due to the fact that nuclear power plants have multiple layers of safety measures to prevent the release of radioactive materials into the environment. The primary source of radioactive wastewater in a nuclear power plant is the cooling water that circulates through the reactor core to remove excess heat. This cooling water can become contaminated with trace amounts of radioactive isotopes, but it is carefully contained and treated within the plant. The treatment of radioactive wastewater in nuclear power plants typically involves filtration, ion exchange, and other purification processes to remove radioactive particles and reduce their concentration to safe levels. The treated water is then monitored for compliance with regulatory standards before being discharged into the environment. However, it is important to note that while nuclear power plants strive to operate safely and minimize any potential releases, accidents or incidents can occur [3,4]. In such cases, there might be a temporary release of radioactive wastewater or other radioactive materials. These situations are considered exceptional and are subject to investigation and regulatory oversight to ensure the protection of public health and the environment. Overall, the goal of a properly functioning nuclear power plant is to maintain the highest levels of safety and containment to prevent the continuous release of radioactive wastewater. Oppositely, the uncontrolled discharge of a large amount of radioactive and highly toxic uranium-containing wastewater without proper treatment will impose a heavy burden to human health and ecosystem stability. Moreover, according to records, there are 4.5 billion tons of uranium in the ocean, which is about 1000 times that of the terrestrial uranium reserves [5]. Therefore, uranium is not extremely rare and is considered to be relatively abundant in the Earth’s crust. It is estimated that the Earth’s crust contains about 2.8 parts per million (ppm) of uranium by weight, which makes it more abundant than elements such as silver, mercury, and platinum. However, when we often refer to uranium as a “rare” resource, it is in the context of its concentrated and economically viable deposits. While uranium is widely distributed in the Earth’s crust, the extraction and processing of uranium ore require significant effort and cost. In fact, the above-mentioned large quantities of uranium in seawater refer to the fact that uranium is dissolved in very low concentrations in the Earth’s oceans. However, extracting uranium from seawater is currently not economically viable on a large scale due to the low concentration levels and the energy-intensive processes required to extract and concentrate the uranium. Most of the world’s uranium production comes from conventional mining operations that extract uranium from high-grade ore deposits. These deposits are relatively rare compared to the overall abundance of uranium in the Earth’s crust. So, while uranium itself is not extremely rare, economically viable and easily accessible sources of concentrated uranium deposits are limited, which affects its availability for various applications, including nuclear power generation. Hence, developing an advanced approach for the extraction of uranium from seawater and radioactive uranium-containing wastewater probably plays a vital role in promoting the development of low-carbon-footprint energy and environmental remediation [6,7].

Based on data from literature [8], the potential for uranium recovery will reach 1158 tU or 3.01 million lb U_3_O_8_ in 2030. The cost of uranium can vary significantly depending on factors such as market conditions, production methods, and geopolitical factors. Historically, the price of uranium (U_3_O_8_) experienced significant fluctuations. At its peak in 2007, the spot price of uranium reached around USD 136 per pound. However, in recent years, the price was relatively low, ranging from about USD 20 to USD 30 per pound [8]. Since 2021, the price started to rise from USD 35.7 per pound to USD 53.6 per pound in 2022, then slightly decreased to USD 50.4 per pound in 2023 (the data come from https://www.focus-economics.com/commodities/energy/uranium/, accessed on 1 July 2023). Due to the high toxicity and radioactivity of uranium, exploring an economical, efficient, and environmentally friendly technology to recover uranium from unconventional resources (such as phosphates) and form wastewater containing uranium is playing a crucial role. Currently, a variety of uranium extraction techniques were proposed, which include adsorption [9], photocatalysis [6], electrocatalysis [10], membrane separation [11], ion exchange [12], and the recently proposed piezoelectric catalysis method [13,14]. However, current research methods mainly focus on synthetic solutions with known matrices [15], while seawater and actual uranium-containing wastewater are complex biogeochemical systems accompanied by all kinds of competing metal ions, organic compounds, high salinity, various microorganisms, and specific pH values [16]. Therefore, the development of uranium extraction strategies adapted to various natural environments became a significant research topic in this field [17,18,19].

In recent years, inspired by the photocatalytic application in heavy metal reduction and environmental remediation [20,21,22,23], photocatalytic uranium reduction technology was favored by many researchers [24,25,26,27,28]. Compared with the above-mentioned methods, photocatalytic reduction has the advantages of rich energy driving force, low cost, high efficiency, and green sustainability [29]. However, the catalytic performance of semiconductor photocatalysts is often limited by the recombination of photogenerated charges, making it difficult to achieve ideal energy utilization efficiency [29]. Therefore, it is particularly important to design superior photocatalysts for reduction through various modification methods to reduce the recombination of photogenerated electrons and holes and therefore improve energy utilization efficiency. 

More recently, research on photocatalytic uranium extraction increased year by year, and more catalysts for uranium extraction and related catalytic mechanisms were discovered. It is well known that photocatalytic technology is a new and useful strategy for uranium extraction from wastewater solution. Photocatalytic uranium extraction from wastewater is a significant development in the field of environmental remediation and resource recovery. The pivotal significance embodies the following aspects, (i) uranium contamination: Uranium is a naturally occurring radioactive element that can be found in various concentrations in the environment, including wastewater. Uranium contamination in water sources can pose a significant threat to human health and the environment. Therefore, photocatalytic technology provides an effective method to remove uranium from wastewater. (ii) Traditional methods for uranium extraction from wastewater involve the use of chemical agents such as precipitation, ion exchange, or adsorption onto solid materials. While these methods can be effective, they often require the use of expensive and environmentally harmful chemicals, and the resulting uranium-laden waste can pose disposal challenges. The relatively simple photocatalytic extraction of uranium offers a more sustainable and environmentally friendly alternative. This process utilizes photocatalysts, typically semiconductor materials, to harness solar energy and activate chemical reactions that break down and remove uranium from wastewater. The photocatalysts can be easily regenerated and reused, making the process more economically viable and reducing waste generation. (iv) One of the major advantages of photocatalytic uranium extraction is that it is a solar-driven process. By utilizing sunlight as the energy source, it eliminates the need for external energy inputs, reducing the overall operational costs and environmental impact of the extraction process. Additionally, the use of renewable solar energy aligns with the goals of sustainable development and carbon neutrality. (v) Uranium is a valuable resource that can be utilized for various purposes, including nuclear power generation and medical applications. Photocatalytic extraction offers the opportunity to recover and recycle uranium from wastewater for resource recovery, thereby reducing the reliance on traditional uranium mining and promoting resource sustainability. (vi) In addition to extracting uranium, photocatalytic processes can also remove other contaminants from wastewater for water purification. The photocatalysts have the ability to degrade organic pollutants and inactivate pathogens, contributing to overall water purification and improving water quality. (vii) The development of photocatalytic uranium extraction from wastewater opens up possibilities for decentralized treatment systems for future implications. It can be applied to various wastewater sources, including industrial effluents and mining runoff, where uranium contamination is a concern. The photocatalytic technology has the potential to mitigate the environmental impacts of uranium mining and processing industries and safeguard water resources. Hence, the significance of photocatalytic uranium extraction lies in its sustainable and solar-driven approach, resource recovery potential, water purification capabilities, and the potential for mitigating uranium contamination in various wastewater sources.

Based on these, this review summarizes the proposed photocatalytic U(VI) reduction mechanism and introduces the identification techniques of uranium species after photocatalytic reduction. The various modification strategies of catalysts for photocatalytic U(VI) reduction are reviewed. The current research progress of photocatalytic U(VI) extraction processes in real uranium-containing wastewater and seawater is evaluated. Finally, the challenges faced by the current photocatalytic uranium extraction process and the prospects for future development are presented.

## 2. Mechanism of Photocatalytic Uranium Reduction and Identification of Reduction Products

The photocatalytic uranium extraction strategy, as an important technology for uranium resource recovery and ecosystem restoration, widely attracted increasing attention. Only in 2022, there were 67 research articles relevant to the photocatalytic reduction of U(VI) (Figure 1A), showing a gradually increasing tendency. Current research mainly focuses on how to improve the catalytic efficiency of photocatalysts, while the forms of uranium and the conversion processes were rarely investigated [30]. In nature, uranium species mainly exist in uranium mine wastewater and seawater in four valence states: III, IV, V, and IV (Figure 1B) [31]. Among them, U(IV) and U(VI) species are the most stable in a natural water environment, and uranyl ions (UO_2_^2+^) usually participate in migration and transformation process [16,32,33]. 

U(VI) can exist as all kinds of U species in aqueous solutions, and the types of U(VI) species change with various acidic and alkaline environments (Figure 1C) [34]. In acidic solutions, U(VI) species mainly exist in the form of UO_2_^2+^. When pH is higher than 4, UO_2_^2+^ species decrease, UO_2_OH^+^, (UO_2_)_3_OH_5_^+^, and (UO_2_)_2_OH_2_^2+^ species gradually increase, leading to a complex distribution of uranium species. The distribution of U(VI) species is greatly influenced by pH. Due to the complexity and diversity of natural water environments, the photocatalysts used for U extraction must have a relatively wide selectivity for different U(VI) species to improve uranium extraction efficiency. Herein, we list the studies on photocatalytic uranium extraction in recent years in Table 1 [35,36,37,38,39,40,41,42,43,44,45,46,47,48,49,50,51,52,53,54,55,56,57,58,59,60,61,62,63,64,65,66,67,68,69,70,71,72].

The main purpose of photocatalytic uranium extraction is to convert soluble U(VI) into insoluble UO_2_ and (UO_2_)(O_2_)·4H_2_O [73] in order to recover them from solution. In general, photocatalytic U(VI) extraction includes the following process in the presence of catalyst. Light is irradiated on the catalyst to generate electron-hole pairs, which are used directly to reduce the soluble U(VI) to insoluble U species. Previous studies showed that uranium reduction may involve three different mechanisms of electron transfer [7,74]. The specific mechanism is shown in Figure 1D.

The first proposed mechanism is a two-step single-electron transfer process, where U(VI) is first reduced to U(V) by accepting an electron (U(VI) + e^−^ → U(V)) and then further reduced to U(IV) by accepting another electron (U(V) + e^−^ → U(IV)) [75,76,77,78]. Furthermore, U(V) can undergo disproportionation under certain conditions to form both U(IV) and U(VI) (U(V) → U(IV) + U(VI)) [69]. 

The second mechanism involves a two-electron reduction process, where U(VI) accepts two electrons and is directly reduced to U(IV) (UO_2_^2+^ + 2e^−^ → UO_2_) [37,79].

The third mechanism involves the reaction of U(VI) with one electron and four protons to produce U(IV) and two water molecules in acidic solution (UO_2_^2+^ + 4H^+^ + e^−^ → U^4+^ + 2H_2_O), or the reaction of U(V) generated from the first mechanism with one electron and four protons to produce U(IV) and two water molecules (UO_2_^+^ + 4H^+^ + e^−^ → U^4+^ + 2H_2_O). In addition, in some specific photocatalytic systems, excess electrons at the conduction band (CB) position will react with dissolved oxygen to generate **·**O_2_^−^ (O_2_ + e^−^ → **·**O_2_^−^), which can also reduce U(VI) via UO_2_^+^ + O_2_^−^ → UO_2_ + 2O_2_ [72].

Obviously, the photocatalytic reduction of U(VI) involves various electron transfer processes. Suitable band structure and high carrier mobility are key factors in the design of photocatalysts. In general, when the redox potential of a target substance falls between the valence band maximum and the CB minimum of a catalyst, the substance may undergo a redox reaction. For the semiconductor itself, a CB potential higher than the U(VI) reduction potential (+0.41 V) is required to facilitate electron transfer [80]. Based on semiconductor photocatalysts, cadmium sulfide (CdS) received wide attention due to its narrow bandgap relative to visible light response and suitable CB position [81]. Qin et al. synthesized NiS@CdS interface Schottky heterojunction photocatalysts with different Ni/Cd molar ratios. The simulated removal efficiency of U(VI) in U-containing wastewater was as high as 99% under 90 min under solar light irradiation. This result is attributed to their good zeta potential and optical properties, narrow bandgap, and the high stability of the composite material [82].

The effective adsorption sites on the photocatalyst surface are very pivotal for photocatalytic uranium reduction, due to these sites determining the interaction between the catalyst and uranium species and the triggering of the reduction reaction [35]. Moreover, the final product of photocatalytic uranium extraction is mainly adsorbed on the surface of the catalyst, hence the type of product U species can be identified by various characterization methods, such as X-ray diffraction (XRD), X-ray photoelectron spectroscopy (XPS), Fourier-transform infrared spectroscopy (FT-IR), transmission electron microscopy (TEM), and X-ray absorption near edge structure (XANES) [29]. For instance, Chen et al. [83] use TT-POR COF-Ni as a photocatalyst to remove uranium (Figure 2A). After photocatalytic reaction, the form of uranium was characterized by XPS, FT-IR, and PXRD. The high-resolution O 1s XPS indicated three different oxygen species, attributable to U-O_peroxo_, U = O_axial_, and H_2_O, respectively (Figure 2B). While the high-resolution U 4f XPS showed a characteristic peak that corresponded to U 4f_7/2_ in UO_2_(O)_2_·2H_2_O at 381.9 eV, implying no change in the valence state of uranium (Figure 2C). Moreover, the PXRD pattern also appeared at some new diffraction peaks at 2θ = 16.8°, 20.2°, 23.4°, and 25.1° (Figure 2D), which can be attributed to metastudtite phase (UO_2_(O)_2_·2H_2_O, and the characteristic peak of UO_2_ at 2θ = 28.1° was not found (Figure 2E). In addition, the appearance of peaks at 902 cm^−1^ assigned to the uranyl group and at 3460 cm^−1^ belonged to O-H vibration from H_2_O in the FTIR spectra after light irradiation (Figure 2F) further corroborated the formation of (UO_2_(O)_2_·2H_2_O.

Additionally, to identify the uranium-containing products of the photocatalytic experiments, U L_III_-edge XANES spectroscopy, Fourier transform extended X-ray absorption fine structure spectroscopy (FT-EXAFS) and wavelet transform (WT) contour maps were measured [84]. The U L_III_-edge XANES spectra (Figure 3(Aa)) of the photocatalyst used were similar to that of the UO_2_ reference sample, indicating the formation of U(IV) species during the photocatalytic reaction. The FT-EXAFS spectra (Figure 3(Ab)) of both compounds showed peaks at ~1.42 Å and 1.97 Å, attributed to the presence of U-O bonds. WT contour plots further confirmed the formation of reduced products (Figure 3(Ac)) and showed a maximum value of 6.2 Å^−1^ in k space and 1.34 Å in R space, well matching with UO_2_ reference powder. When the immobilized TiO_2_ nanotube arrays (TNAs) were used as photocatalyst to extract and recover uranium from solution [25], the flake-like structure on the catalyst surface was observed by TEM image (Figure 3(Ba) left), and the EDS (Figure 3(Ba) middle, right) elemental mappings confirmed a relatively uniform distribution of U and O on catalyst. In addition, crystal lattice fringes of 0.257 nm and 0.182 nm, which are attributable to the (210) and (163) lattice planes of UO_3_ and UO_3_·H_2_O, respectively, were visible from high-resolution TEM images (Figure 3(Bb,Bc)). After photocatalytic reaction, recovered U product on filter paper was orange yellow (Figure 3(Bd)).

## 3. Photocatalysts for Uranium Recovery from Solution

In the past decade, photocatalytic uranium extraction was regarded as a simple, efficient, and environmentally friendly method to solve U(VI) contamination in water. In general, TiO_2_ is the most widely used photocatalytic material in recent years for solving U(VI) pollution problems [24,37,72,85,86,87,88,89]. However, TiO_2_ as a photocatalyst indicated some potential disadvantages for the removal of U(VI): (1) recombination of photogenerated electron-hole pairs in TiO_2_ leads to the decline in photocatalytic efficiency [90]; (2) an artificial UV light source is required, and it is difficult to achieve U(VI) removal using sunlight, which demands high energy consumption [91]. With development of hybrid nanocomposite and heterojunction-structured catalysts, the separation efficiency of photogenerated carriers was remarkably improved and therefore promoted photocatalytic performance [36,54,79]. In addition, the surface structure of the photocatalyst itself including defects, vacancies, and heteroatom doping often influences catalytic activity. 

### 3.1. Semiconductor Heterojunction Structure

A heterojunction represents the interface between two different semiconductors with unequal band structure, which can result in band alignments [92]. The energy level difference in semiconductor heterojunction drives the effective separation and migration of photogenerated charges, and therefore improves the photocatalytic ability for water pollution treatment and the degradation of organic pollutants [92,93]. In recent years, heterojunction materials were widely used in the photocatalytic extraction of U(VI) [94,95].

According to the relative positions of the band structures of photocatalysts, binary heterojunction photocatalysts can be divided into three types, type I (straddling gap alignment, Figure 4A), type II (staggered gap alignment, Figure 4B), and type III (broken gap alignment, Figure 4C) [80]. Among them, the II-type heterojunction attracted extensive attention due to its unique band structure arrangement, ensuring effective separation of photogenerated charge carriers [96,97]. For example, g-C_3_N_4_/TiO_2_ binary heterojunction showed high U(VI) reduction ability compared with single g-C_3_N_4_ or TiO_2_ during photocatalytic extraction of uranium [54]. The g-C_3_N_4_/TiO_2_ with II-type heterojunction arrangement has a lower recombination rate of photogenerated electrons and holes and higher charge transfer efficiency, which was confirmed by the photoluminescence spectroscopy (PL), photocurrent response, and electrochemical impedance spectroscopy, revealing the origin of the enhancement of the photocatalytic activity (Figure 5A–D). For a niobate/titanate (Nb/TiNFs) heterojunction prepared using a one-step hydrothermal method [44], electrons generated on titanate migrate to the CB of niobate due to their CB offset, which greatly suppresses the recombination of electron-hole pairs. The I-type heterojunction structure of the titanate and niobate composite materials compensates for the low photocatalytic activity of the monomer materials in U(VI) extraction (Figure 5E,F). Moreover, composite materials, such as MoS_2_@TiO_2_ [52], MoS_2_@g-C_3_N_4_ [94], ZnS@g-C_3_N_4_ [98], CuS/TiO_2_, and so on, were used as excellent photocatalysts for U(VI) reduction. 

The direct Z-scheme heterojunction and type II heterojunction have a similar energy band structure arrangement, but the direct Z-scheme heterojunction involves the recombination of electrons from the CB of semiconductor A and holes from the valence band of semiconductor B (Figure 4D). This unique internal charge transfer mechanism allows the direct Z-scheme heterojunction to maintain both high oxidation and reduction abilities. Liu et al. [99] recently reported a ZnS/WO_3_ Z-type heterojunction photocatalyst with excellent visible light photocatalytic reduction performance for U(VI). When the two materials come into contact, electron transfer causes negative charge accumulation on the WO_3_ side and consumption of negative charge on the ZnS side, resulting in the formation of an internal electric field at the material interface (Figure 6A). The internal electric field can serve as a one-way electron channel from WO_3_ to ZnS. Under illumination, the accumulated e^−^ in the CB of WO_3_ migrates to the contact interface and recombines with the photo-induced h^+^ in the VB of ZnS, resulting in a long separation lifetime of e^−^ (Figure 6B). This greatly enhances the photocatalytic performance of the catalyst. 

Note that when a metal/semiconductor forms a Schottky junction, the metal will accumulate a large number of electrons or holes. The resulting Schottky barrier will prevent electrons and holes from transferring to the semiconductor, ensuring single one-way transfer of electrons and holes, promoting the separation of photo-generated electrons and holes, and correspondingly improving the photocatalytic performance. Qin et al. [82] synthesized a NiS/CdS composite photocatalyst and demonstrated the existence of an interface Schottky junction between CdS and NiS, boosting spatial charge separation through density functional theory (DFT) calculations for highly efficient photocatalytic reduction in U(VI) (Figure 6C–F). The resultant U(VI) extraction efficiency reached 99% within 90 min from the uranium-containing solution. Dai et al. [100] first synthesized a magnetic graphene oxide-modified graphitic carbon nitride (mGO/g-C_3_N_4_) nanocomposite material, which can effectively catalyze the reduction in U(VI) from wastewater under visible light irradiation from LEDs, achieving a U(VI) extraction efficiency as high as 96.02%. Wan et al. [101] used S-injected engineering to functionalize the atoms into the TiO_2_/N-doped hollow carbon sphere (TiO_2_/NHCS) heterostructure to form Schottky junctions for spatially separating photogenerated electrons and optimizing the TiO_2_ energy level structure. The results show that the reduction efficiency of U(VI) within 20 min exceeded 90% and the removal rate per unit mass reached 448 mg g^−1^. 

### 3.2. Defective Semiconductors

In photocatalysis, defect is a vital parameter to be pre-considered for the design of photocatalysts and catalytic performance. Crystallographic defects, which often occur at where the perfect periodic arrangement of atoms or molecules in the crystalline materials is disrupted or broken, inevitably lead to the imperfection of perfect crystal structure and therefore widely exist in all photocatalytic materials and increase active sites and greatly promote photocatalytic performance [102]. For semiconductors, the band structure can be improved by adjusting the defects, and thereby improving their light absorption capacity [29]. In addition, surface defect centers can also act as reactive sites, increasing the effectively active surface area. As one of the most common defects in semiconductor materials, vacancies can improve the photocatalytic performance by adjusting the electronic structure and light absorption capability [103,104]; for example, oxygen vacancy-rich tungsten oxide nanowires (WO_3-x_) have a narrower bandgap energy and higher charge carrier separation efficiency compared to regular WO_3_ [105]. DFT calculations further demonstrated that the introduction of oxygen vacancies (OVs) in WO_3-x_ resulted in the spin-polarized state of W 3d electrons in WO_3-x_, greatly suppressing the recombination of photogenerated electrons and holes (Figure 7A). Compared to WO_3_, WO_3-x_ exhibited higher photocatalytic performance in U(IV) reduction. Hydrothermal reaction of a carbonized bacterial cellulose (BC) with thiourea and sodium molybdate dihydrate afforded a heterojunction-structured BC-MoS_2_, followed by H_2_/Ar mixed plasma treatment to generate S-vacancies (SVs) in MoS_2_ and form BC-MoS_2-x_ heterojunction containing Schottky junction and SVs, which were used as photocatalysts for efficient uranium reduction. The resultant removal rate of uranium over a wide range of U(VI) concentrations was up to 91% [33]. Li et al. constructed an oxygen vacancy (OVs)-rich g-C_3_N_4_-CeO_2-x_ heterojunction [106], kinetic characterization, and DFT calculations show that photogenerated electrons were transferred from g-C_3_N_4_ to a CeO_2-x_ heterojunction through the built-in electric field generated by the heterojunction and were captured by shallow traps created by surface vacancies, and thereby achieving spatial separation (Figure 7B). The separation rate and lifetime of photoinduced carriers were significantly increased, and photocatalytic activity for U(VI) reduction was significantly enhanced (39 times higher than that of g-C_3_N_4_). 

### 3.3. Elemental Doping of Semiconductors

Doping can modify the surface structure, light absorption, defects, charge density, and carrier separation efficiency of photocatalytic materials [107]. In recent years, element-doped materials received extensive attention in the field of photocatalytic U(VI) reduction. As shown in Figure 8A, Wang et al. [108] used non-metallic-doped photocatalysts and carbon-doped boron nitride (BN) BCN nanosheets for photocatalytic uranium reduction. By optimizing composite, calculations of energy gap and density of state of BCN, and experimental date, the results show that the doping in the form of carbon rings could regulate the bandgap and electronic structure by manipulating carbon amount. The introduction of carbon rings improved the surface structure and light absorption capacity of the catalyst, and the extraction rate of U(VI) reached 97.4% under visible light irradiation for 1.5 h. For the Ag-doped CdSe (Ag-CdSe) nanosheet, the doping of Ag resulted in an increase in the density of photogenerated carriers and a narrowed bandgap, while also suppressing the recombination of photogenerated electrons and holes [109]; 3% Ag-CdSe nanosheets reached a removal rate of U(VI) of 96%, which is 1.9 times higher than that of the original CdSe nanosheets (50%). As shown in Figure 8B, to achieve higher photocatalytic efficiency in the extraction of U(VI), up-conversion Er-doped ZnO nanosheets were used as a photocatalyst [110]. Er doping induced up-conversion properties and suppressed recombination of photogenerated carriers, therefore enhancing light absorption capacity and accelerating U(VI) extraction. At an initial U(VI) concentration of 200 mg L^−1^, 4% Er-doped ZnO reached a high extraction efficiency of uranium at about 91.8% within 3 min. 

### 3.4. Other Strategies

The band structure, light absorption, and carrier separation efficiency of photocatalysts are some key factors in affecting the photocatalytic performance. However, since the photocatalytic extraction of U(VI) occurs on the surface of the catalyst, optimizing the surface structure of the catalyst is another important method for improving photocatalytic performance. To further enhance the U(VI) extraction ability of carbon nitride photocatalysts, a one-step molten salt method was adopted to prepare a new bifunctional carbon nitride material (CN550) (Figure 9A) [111]. Compared with g-C_3_N_4_, CN550 has a high surface area, good adsorption capacity, and a high photocatalytic activity. More recently, various porous framework materials, such as metal–organic frameworks (MOFs), covalent organic frameworks (COFs), and hydrogen-bonded organic frameworks (HOFs), were regarded as the state-of-the-art materials for photocatalytic uranium extraction due to their high surface area and more active sites. Li et al. [27] proposed a new uranium extraction strategy based on post-synthetically functionalized MOF PCN-222, which is an extremely robust MOF, composed of Zr_6_(μ_3_-O)_4_(μ_3_-OH)_4_(H_2_O)_4_(OH)_4_ clusters and photoactive meso-tetra(4-carboxyphenyl)porphyrin (TCPP) linkers. PCN-222 was treated with aminomethylphosphonic acid and ethanephosphonic acid to afford PN-PCN-222 and P-PCN-222, respectively (Figure 9(Ba)). The uranium uptake results show that PN-PCN-222 can completely remove uranyl ions in an extremely wide uranyl concentration range in solution and reach a very high absorption capacity at about 1289 mg g^−1^, breaking the maximum adsorption capacity of previously reported MOF materials (Figure 9(Bb)). Under visible light irradiation, the photo-induced electrons in the PN-PCN-222 matrix reduce the U(VI) pre-enriched in PN-PCN-222, producing neutral uranium species that disperse the MOF structure, exposing more active sites to capture more U(VI) (Figure 9(Bc)). Overall, the development of photocatalytic strategies for efficient U(VI) extraction is recognized as an efficient, environmentally friendly, and cost-effective approach, while rational design and optimization of photocatalysts made significant progress in photocatalytic technology. At present, many articles describe the design of photocatalysts and the principles of photocatalytic systems in detail, and researchers can obtain more inspiration from them [29,112,113].

## 4. Photocatalytic U(VI) Extraction from Uranium-Containing Wastewater and Seawater

Unlike seawater and groundwater, the composition of actual uranium mining wastewater is very complex. It often contains various metal salts, organic matter, and microorganisms [114,115]. Although some advanced photocatalysts indicated good selectivity for the photocatalytic reduction in U(VI) in the presence of multiple ions and organic compounds, the concentrations of interfering ions and organic compounds in the simulated uranium-containing wastewater used in the laboratory are at the milligram level. In the actual uranium-containing wastewater, the concentrations of interfering ions and organic compounds are thousands or even tens of thousands of times higher than those of the simulated uranium-containing wastewater. The development of highly efficient photocatalysts for the extraction of U(VI) from actual uranium mining wastewater became a challenging research topic. To develop an efficient photocatalyst for U(VI) extraction from rare earth tailings wastewater, Liu et al. constructed a SnS_2_-covalent organic framework van der Waals heterojunction (SnS_2_COF) (Figure 10(Aa)) [28]. Experimental results show that the removal rate of U(VI) by SnS_2_COF reached 1123.3 mg g^−1^ under the conditions without a protective atmosphere, and the removal rate of U(VI) in rare earth tailings wastewater reached 98.5% (Figure 10(Ab,Ac)). This provides a designable approach to address uranium pollution problems in real water environments.

The concentration of uranium in seawater is about 3.3 μg·L^−1^ [16]. Although this concentration is very low, the total uranium content in seawater is 4.5 billion tons, which is approximately 1000 times that of the uranium reserves on land [116]. Given the rapid development of nuclear energy, uranium reserves in seawater alone are sufficient to sustain nuclear energy consumption for thousands of years even without considering any recycling of radioactive waste [117]. However, due to the extremely low uranium concentration in natural seawater and the presence of potential competing ions, it is still challenging to develop efficient photocatalytic technology for uranium extraction from seawater [118]. To realize the photocatalytic uranium extraction from seawater, Yu et al. designed and synthesized two new donor–acceptor conjugated microporous polymers (CMPs), ECUT-CO and ECUT-SO with perylene as donor and alternating acceptors (9H-fluoren-9-one and dibenzo [b,d] thiophene 5,5-dioxide) (Figure 10(Ba)) [6]. Nitrogen adsorption isotherms and pore size distribution curves were used to demonstrate the microporous structural characteristics of all CMPs (Figure 10(Bb,Bc)), and the larger specific surface area and pore structure provide more reactive sites for photocatalytic reactions. The experiments of extracting U(VI) from real seawater showed that uranium in seawater can be completely removed within 60 min, and the photocatalytic uranium reduction activity of ECUT-SO did not decrease significantly after three cycles, indicating the excellent practical value (Figure 10(Bd–Bf)). Although significant contributions were made in recent years to the development of photocatalytic materials for uranium extraction from seawater, it remains a challenge to develop a comprehensive photocatalytic system that meets the requirements for uranium extraction from seawater [119,120]. To develop more efficient photocatalytic materials for uranium extraction from seawater, the selective adsorption ability of catalysts for uranium in seawater should be regarded as an important factor. According to current studies, porous organic framework materials such as COFs and MOFs exhibit excellent adsorption capacity and anti-biofouling ability for photocatalytic uranium extraction in seawater, and can maintain a good U(VI) reduction effect in seawater [113,121,122]. 

Although uranium extraction from seawater and uranium-containing wastewater by photocatalytic reduction made some great progress, the extremely low uranium concentration in seawater and uranium-containing wastewater poses a major challenge to the extraction process. To effectively design the photocatalytic U(VI) reduction process, both the initial concentration of uranium and the amount of photocatalyst must be considered. Li et al. investigated the influence of different initial U(VI) concentrations on U(VI) reduction efficiency and found that with increasing the initial concentration of U(VI), the adsorption amount during the dark reaction remained almost constant and the rates of the photocatalytic reactions became similar, implying the direct correlation between adsorption amount and photocatalytic kinetics processes [37]. To investigate the effect of catalyst dosage on the performance of photocatalytic U(VI) reduction, Liang et al. studied the U(VI) reduction performance with different catalyst dosages [123]. The results show that the U(VI) reduction efficiency decreased when the catalyst dosage exceeded a certain threshold (> 0.4 g/L), revealing a shielding effect, where an excess of suspended catalyst reduced the penetration of light in solution, thereby reducing the light utilization efficiency. Especially for nanocatalysts, it is often necessary to determine an optimal catalyst dosage to avoid a decline in catalytic performance, due to the shielding effect. Additionally, the initial concentration of U(VI) also needs to be further investigated. Note that too low a concentration is not conducive to the capture of the catalyst, while too high a concentration may result in the slow release of U(VI) photocatalytic reduction products adsorbed on the catalyst surface, thereby hindering the exposure of active sites. Hence, photocatalytic uranium extraction technology can be used for all wastewater containing uranium, no limitation is required for the concentration range of uranium in wastewater, and only uranium extraction efficiency will be influenced, indicating a poor or an excellent extraction efficiency for uranium.

## 5. Current Challenges and Perspective

Although significant progress was made in photocatalytic uranium extraction technology in recent years, the commercialization of the results is still in its infancy. More breakthroughs are needed to develop a green, energy-saving, and environmentally friendly comprehensive system for photocatalytic uranium extraction. Herein, the current challenges and the future development direction from the perspectives of catalysts, environmental systems, and catalytic processes are discussed.

### 5.1. Recovery and Regeneration of Catalysts

The performance evaluation of uranium extraction by photocatalysis is usually based on kinetic models of reaction time and removal rate [51,94,120,124,125]. However, the kinetics of U(VI) extraction is not the only criterion for performance evaluation in practical processes. The recyclability and reuse of catalysts should also be considered. According to current reports, the catalysts used in photocatalysis are usually powder materials [70,126,127]. When these powder materials are dispersed into the water environment, they may cause secondary pollution, which greatly limits the practical application of photocatalytic technology. The separation and recovery of these powders became a major challenge. The current separation methods include centrifugal separation, membrane separation, etc., but these separation and recovery methods are expensive and have strict conditions, which are not conducive to popularization. Therefore, it is necessary to develop efficient, pollution-free, and easy-to-recycle photocatalysts, and to study the separation and recovery methods of powdered catalysts in water environments. Li et al. [35] developed a magnetically collectable TiO_2_/Fe_3_O_4_ and its graphene composite, and systematically studied the effects of initial uranium concentration, pH of solution, ion strength, wastewater composition, and organic pollutants on the removal of U(VI) by TiO_2_/Fe_3_O_4_. This provides a new design concept for the separation and recovery of uranium from radioactive wastewater. 

Generally, the photo-reduced U(IV) is adsorbed on the surface of photocatalysts, which is detrimental to the exposure and sustainability of active sites [35,111,123,128]. To maintain the recycling performance of photocatalysts, catalyst regeneration technology must be used to desorb U(IV) adsorbed on the surface of the photocatalysts. Currently, the most commonly used catalyst regeneration methods include acid washing [27,50,111], carbonate solution washing [13,31,51,129], and air re-oxidation [130]. Although these methods can restore some catalytic performance, there are still some issues. Pickling requires the use of strong acids, which is expensive and causes environmental pollution, while carbonate washing and air re-oxidation are time-consuming. In addition, the performance of the catalysts declines significantly after repeated use. Therefore, it is of great significance to develop environmentally friendly, efficient, and sustainable methods for catalyst regeneration.

### 5.2. Photocatalytic U(VI) Extraction under Sunlight

Currently, various photocatalytic semiconductor materials were reported for photocatalytic U(VI) extraction. However, most of these materials only exhibit some photocatalytic U(VI) extraction ability under simulated light sources under laboratory conditions, which is uneconomical and environmentally unfriendly. The best energy source for photocatalysis should be real sunlight, not simulated light sources in the laboratory. Unfortunately, the actual sunlight has long wavelengths, and the short wavelengths (ultraviolet light) account for less than 10% of the energy spectrum, severely limiting the efficiency of photocatalysis. In addition, the intensity of sunlight varies with weather and seasons, which has a significant impact on the photocatalytic performance of the catalyst. Therefore, the development of semiconductor catalysts that can achieve full-wavelength light absorption is of great significance for the photocatalytic extraction of U(VI) under sunlight. Elemental doping, defect engineering, and heterojunctions can significantly enhance the light absorption ability of semiconductors and suppress the recombination of photogenerated holes and electrons, providing a good strategy for catalyst design. 

### 5.3. Coupling of Photocatalytic Technology with Other Techniques

Traditional photocatalytic technologies face challenges such as weak light absorption ability, recombination of photogenerated charge carriers, and difficulties in catalyst recovery. As we summarized before, various attempts were made to address these issues. If photocatalysis is combined with other catalytic technologies, can it overcome some limitations in the photocatalytic process and improve the efficiency? For instance, Dai et al. [131] fabricated a g-C_3_N_4_/Sn_3_O_4_/Ni (CSN) electrode supported on nickel (Ni) foam and employed it as an anode material for photoelectrochemical U(VI) reduction (Figure 11A). Due to the excellent light absorption ability and enhanced charge carrier transfer efficiency of the g-C_3_N_4_/Sn_3_O_4_ heterostructure, the photocatalytic U(VI) reduction in air by the CSN electrode is as high as 94.28%. Furthermore, the mechanistic analysis revealed that the exceptional photoelectrocatalytic performance of the CSN electrode towards U(VI) was mainly attributed to its enhanced visible light absorption, facilitated charge carrier separation of photo-generated electrons and holes, and reduced bandgap caused by the combination of the g-C_3_N_4_/Sn_3_O_4_ heterojunction with an applied voltage. Due to the spatial separation of the anode and cathode in the photoelectrochemical cell, the photocathodic reduction process and photoanodic oxidation process are separately conducted, thereby minimizing the interference of reactive oxygen species (ROS) generated by photo-generated electrons and holes. In addition, U(VI) was reduced to U(IV) and immobilized on the cathode surface. When the catalyst is used as an anode material, the reusability and operational lifespan of the catalyst can be greatly improved. Despite reports demonstrating the relative advantages of photoelectrocatalysis over conventional photocatalysis in U(VI) extraction, the development of anode catalysts is significantly constrained and the understanding of the catalytic mechanism remains limited. This field requires further research.

In recent years, piezocatalysis emerged as a promising approach for the efficient conversion of mechanical energy, demonstrating potential applications in energy crises [133], environmental remediation [134], biomass conversion [135], CO_2_ reduction [136], and water splitting [137]. Photocatalysis, as an advanced oxidation technology, is widely employed for efficient pollution control. However, the purification efficiency of environmental photocatalysis is constrained by the rapid recombination of photogenerated electron-hole pairs. Therefore, it is proposed to generate an internal built-in electric field through the piezoelectric effect to enhance the separation efficiency of photogenerated charge carriers and achieve better photocatalytic performance [132] (Figure 11B). For example, Yuan et al. [138] synthesized chlorine-doped ZnO nanotubes and first proposed the adjustment of piezoelectric properties to enhance the coupling system in photocatalytic degradation of organic dyes. The migration process of charge carriers is optimized by utilizing the radial piezoelectric field and the high-speed electron flow along the axial direction. Meanwhile, Chen et al. [139] achieved efficient recovery of gold from thiosulfate solution by employing defect-rich MoS_2_ nanoflowers (DR-MoS_2_ NFs) as a piezo-photocatalyst for the reduction in Au(I). Due to the reduced Schottky barrier at the interface and enhanced piezoelectric potential, DR-MoS_2_ NFs enable rapid separation of photoinduced charge carriers, resulting in ultrafast reduction in Au(I) under indoor light irradiation with the assistance of ultrasound treatment. Although some recent studies demonstrated the superior performance of piezo-photocatalytic coupling systems compared to traditional photocatalysis, there is still a significant knowledge gap in the rational application of piezoelectric photocatalysis for environmental remediation [132]. Particularly, there is a lack of relevant research on piezo-photocatalytic U(VI) extraction. Furthermore, whether it is a stand-alone photocatalytic system or a coupled system with other catalytic technologies, the current major challenges lie in the deployment of catalytic systems and the stability of catalysts.

### 5.4. Engineering Aspects of Scaling up Experiments in Real Water Environments

The ultimate goal of photocatalytic uranium extraction is to solve the engineering challenges of large-scale uranium extraction in real water environments and achieve stable and economically viable uranium recovery. Up to now, although there were successful cases of uranium extraction from real uranium mine wastewater and seawater, there are still many bottlenecks in expanding the experimental scale in real water environments. Substantial breakthroughs are still needed to address these challenges. Before carrying out the uranium extraction engineering in the actual water environment, it is crucial to have a comprehensive understanding of the water environment, because there is a significant difference between the real water environment and the laboratory-simulated environment. It would be valuable to study the solvation structure of dominant uranium species and the interactions between uranyl complexes and ligands in real water environments with specific pH and high salinity. At the same time, the effects of water temperature, flow rate, and other coexisting pollutants need to be evaluated in order to determine the optimal location and conditions for conducting experiments. Specifically, water pollution is a major and unavoidable challenge for engineering experiments in real water environments. Carious competing ions, microorganisms, and organic pollutants exist in real water environments, which seriously affects the efficiency of uranium extraction and catalyst regeneration. In this sense, an in-depth study of these effects is necessary if large-scale uranium extraction experiments are to be performed in real water environments.

Another major challenge comes from the deployment of photocatalytic systems. In practical industrial applications, the selection of appropriate reactor type and size can ensure efficient photocatalytic reactions and maximize the utilization of light energy, which is crucial to obtain cost-effective uranium products. In addition, the light source and lighting conditions are crucial for exciting photocatalytic reactions and important parameters in the process conditions. In a real water environment, immobilizing the catalyst can improve its reusability and prevent secondary pollution caused by the catalyst. Engineering issues require a comprehensive consideration of knowledge in various fields, such as materials science, chemical engineering, and environmental engineering to achieve the feasibility and effectiveness of photocatalytic technology in practical applications. Comprehensive assessments of these aspects were not conducted in previous research reports. Therefore, further efforts are needed to facilitate the deployment of large-scale photocatalytic systems.

### 5.5. Cost Evaluation

Notably, photocatalytic uranium extraction methods were never fully demonstrated on a commercial scale. Therefore, there are certain risks in the scaling up and commercialization of this technology. This review provides a preliminary cost estimation comparison for photocatalysis, ion exchange, and solvent extraction methods. It is worth noting that the solvent extraction method was successfully commercialized, with only an estimated operating cost of USD 32 per pound for U_3_O_8_. Additionally, according to literature reports, the preliminary estimated operating cost of the ion exchange method is USD 33–54 per pound of U_3_O_8_ [140]. If photocatalytic uranium extraction is conducted from seawater with a uranium concentration of approximately 3.3 mg/t, it would require extracting 1 pound for U_3_O_8_ from approximately 137,452 t of seawater [16]. Assuming a catalyst price of USD 10 per kilogram, considering the regeneration and consumption of the catalyst, as well as the layout of the catalytic system and equipment loss, the operating cost of photocatalytic uranium extraction needs to be kept within USD 33 per pound for U_3_O_8_. Moreover, the price of uranium must remain stable at or above USD 44–61 to meet the economic requirements of this investment. The total cost per unit is affected by the uranium concentration in the seawater. Every 10% (or 20%) decrease in uranium concentration increases the overall cost by USD 2–3 (or USD 5–6) per pound for U_3_O_8_. Conversely, a 10% (or 20%) increase in uranium concentration reduces the total unit cost by USD 1–2 (or USD 3–4) per pound for U_3_O_8_ [140].

Hence, cost-effectiveness is a key factor in realizing the large-scale industrial application of photocatalytic U(VI) extraction technology. However, current studies lack a fair comparison of the cost-effectiveness analysis of different catalytic processes and catalysts. It is recommended that future research on photocatalytic U(VI) extraction includes an associated economic evaluation of the catalysts used and catalytic process. Although these cost estimates may differ from industrial-scale applications, they can still reflect the application potential of a catalyst or catalytic process, or provide a reasonable basis for comparison with other studies. This review recommends a cost–benefit analysis based on: (1) synthesis of the catalyst; (2) deployment of the catalytic system; (3) external energy required; (4) consumption during the catalytic process; and (5) catalyst regeneration.

## 6. Conclusions

Significant progress was made in photocatalytic U(VI) extraction technology. Summarizing the existing work will help to overcome technical bottleneck and deepen the understanding of photocatalytic technology, thereby promoting the development of photocatalytic U(VI) extraction technology. This article provides an overview of the current research status of photocatalytic U(VI) extraction technology, including the reaction mechanism, catalyst design, and research progress in the extraction of uranium from real water environments. Furthermore, the main challenges encountered in photocatalytic U(VI) extraction and future directions in this field are highlighted. We hope this review can serve as inspiration for researchers working in this field and contribute to the advancement of this field.

## Figures and Tables

**Figure 1 nanomaterials-13-02005-f001:**
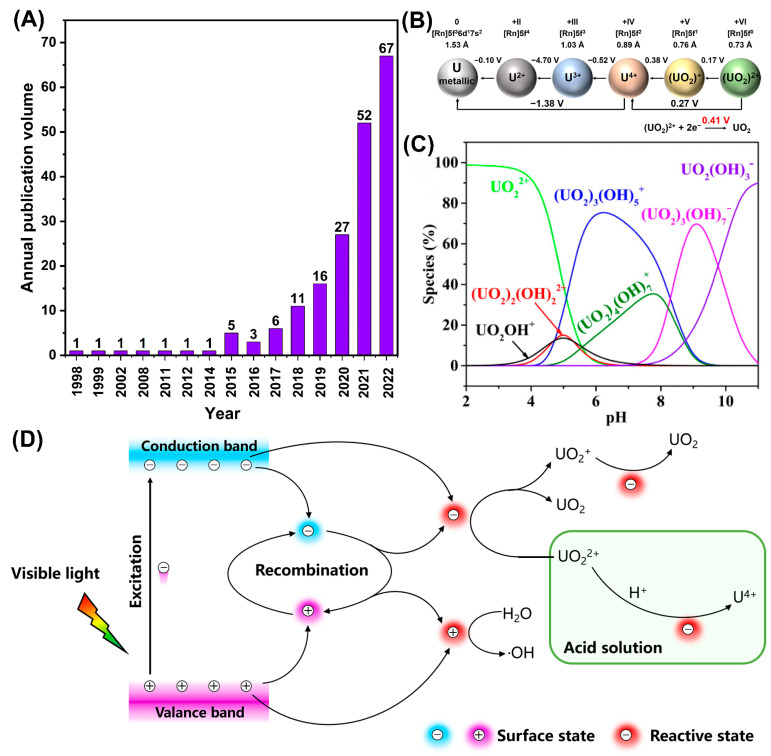
(**A**) Research on photocatalytic U(VI) reduction. These data are based on the search results from “Web of Knowledge” using the keywords “photocatalytic reduction of U(VI)”. (**B**) Chemical oxidation states of uranium. (**C**) The forms of uranium species under different pH conditions. Reproduced with permission [34]. Copyright (2017) Elsevier. (**D**) The photocatalytic U(VI) reduction mechanism.

**Figure 2 nanomaterials-13-02005-f002:**
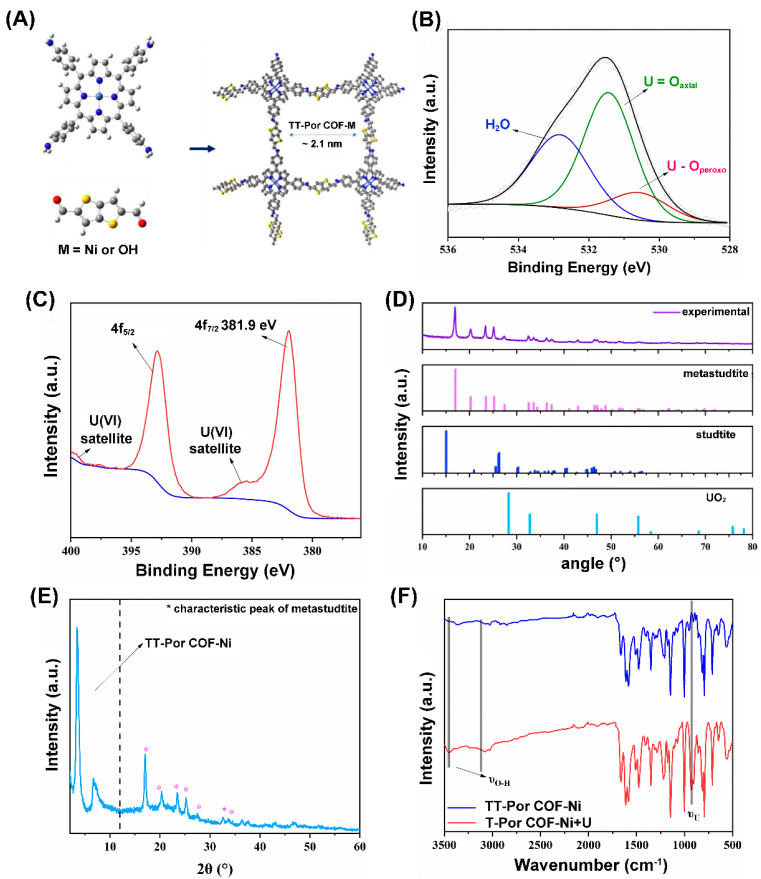
(**A**) Synthesis of the TT-Por COF-Ni. (**B**) The O 1s high-resolution spectra of O 1s, (**C**) and U 4f. (**D**) XRD patterns after irradiation. (**E**) PXRD after illumination. (**F**) FT-IR after illumination (red line) and before illumination (blue line). Reproduced with permission [83]. Copyright (2023) Elsevier.

**Figure 3 nanomaterials-13-02005-f003:**
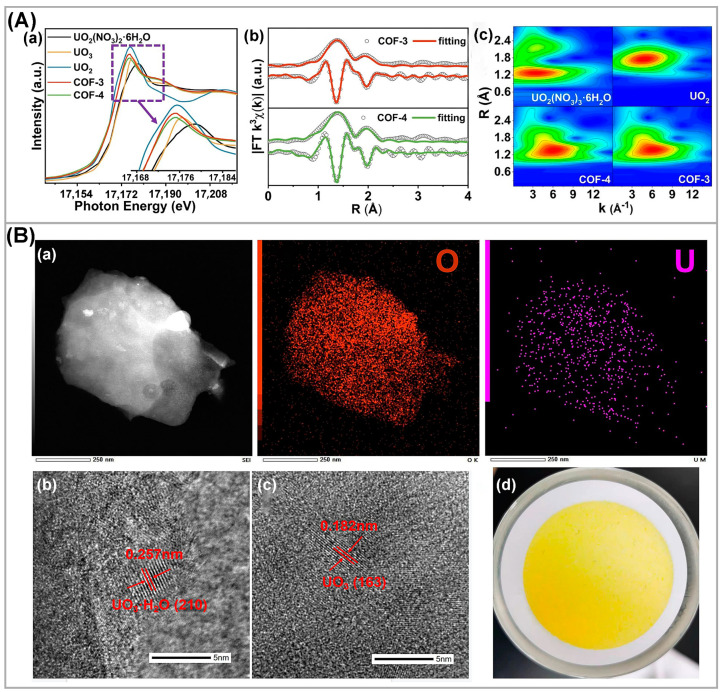
(**Aa**) U L_III_-edge XANES spectra for COF-3 and COF-4 after photocatalysis. UO_2_(NO_3_)_2_·6H_2_O, UO_2_, and UO_3_ are employed for comparison. (**Ab**) EXAFS fitting curve for COF-3 and COF-4 after uranium extraction studies. (**Ac**) WT contour plots for COF-3 and COF-4. Reproduced with permission [84]. Copyright (2023) Springer Nature. (**Ba**) EDS images of uranium recovery products; (**Bb**,**Bc**) HRTEM images of uranium recovery products; (**Bd**) U recovery product filter cake. Reproduced with permission [25]. Copyright (2023) Elsevier.

**Figure 4 nanomaterials-13-02005-f004:**
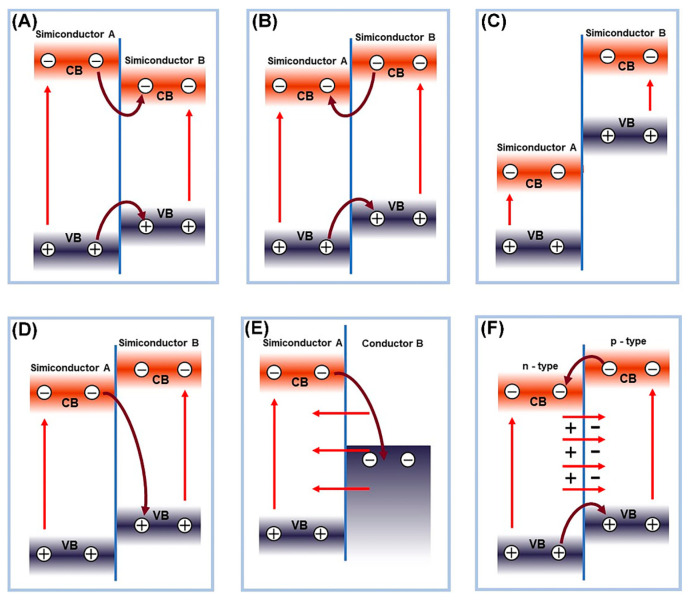
Schematic diagram of electron-hole separation in different types of heterojunctions, (**A**) type-I heterojunction, (**B**) type-II heterojunction, (**C**) type-III heterojunction, (**D**) Z-scheme heterojunction, (**E**) Schottky heterojunction, and (**F**) p-n heterojunction.

**Figure 5 nanomaterials-13-02005-f005:**
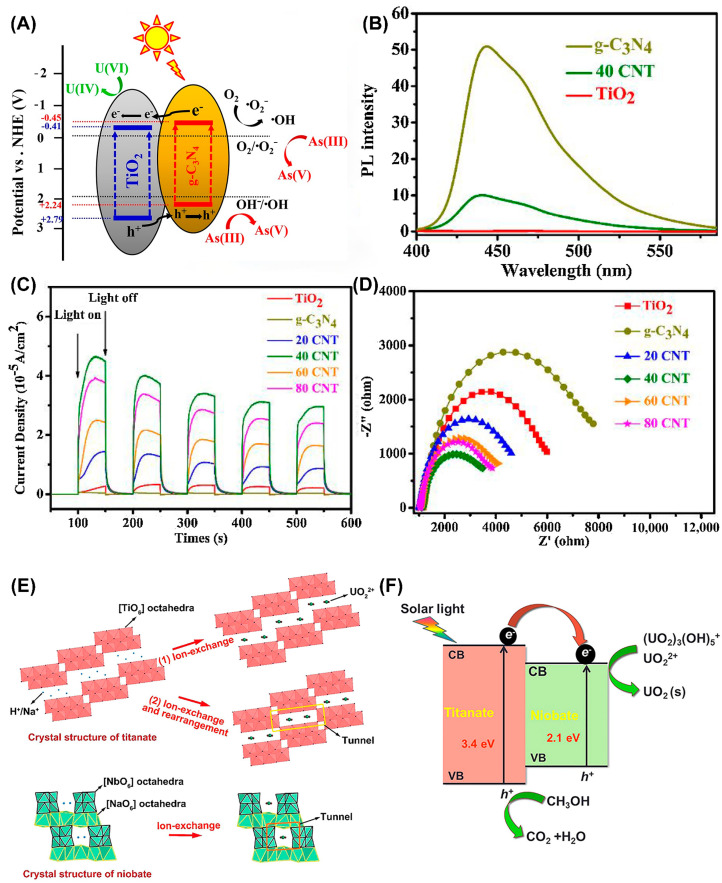
(**A**) Electron transfer mechanism in g-C_3_N_4_/TiO_2_ type-II heterojunction; (**B**) PL spectra, (**C**) transient photocurrent responses, (**D**) and EIS curves. Reproduced with permission [54]. Copyright (2018) Elsevier. Schematic representation of (**E**) adsorption model, and (**F**) photocatalytic reduction in U(VI) mechanism by Nb/TiNFs. Reproduced with permission [44]. Copyright (2018) Elsevier.

**Figure 6 nanomaterials-13-02005-f006:**
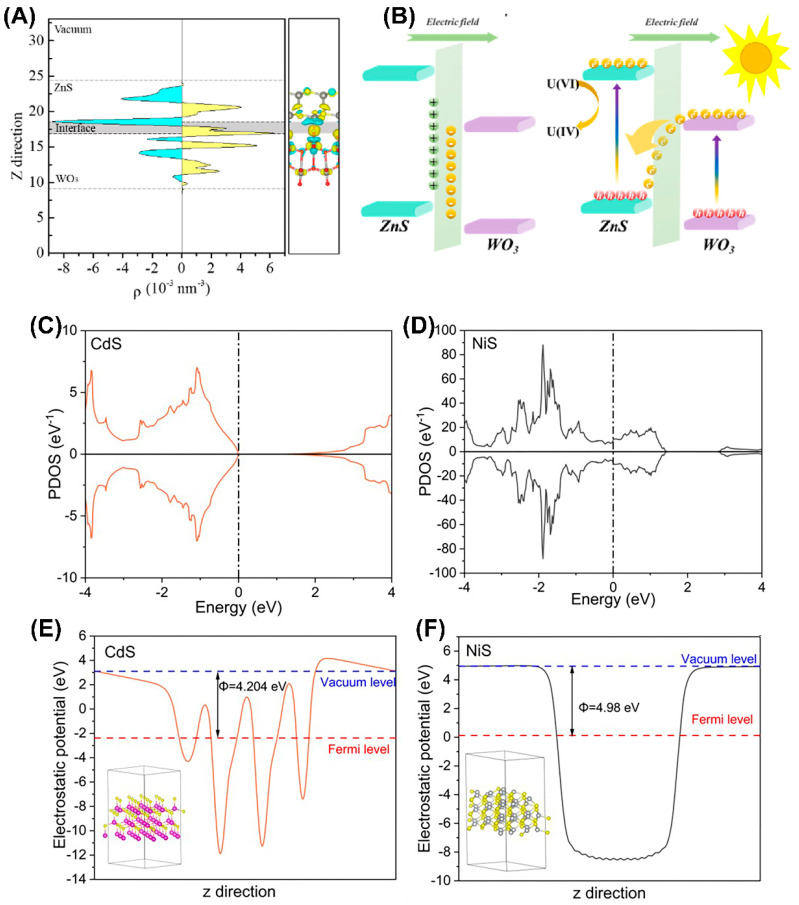
(**A**) The plane-averaged charge density difference and 3D charge density difference, (**B**) and electron transfer mechanism of Z-scheme heterojunction for photocatalytic U(VI) reduction. Reproduced with permission [99]. Copyright (2023) Elsevier. The calculated local density of states (DOS) for (**C**) CdS and (**D**) NiS; calculated electrostatic potentials of the (**E**) CdS (001) facet and (**F**) NiS (102) facet. Reproduced with permission [82]. Copyright (2023) Elsevier.

**Figure 7 nanomaterials-13-02005-f007:**
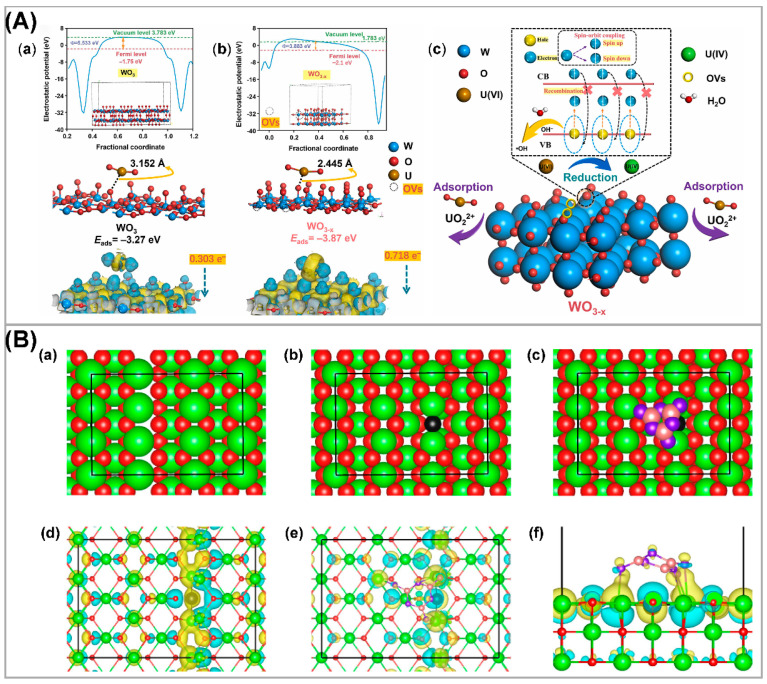
(**A**) The work function, DFT calculations on adsorption of UO_2_^2+^ into materials and EDD image for (001) plane of (**a**) WO_3_ and (**b**) WO_3-x_. (**c**) The proposed mechanism of U(VI) adsorption and photoreduction over the WO_3-x_. Reproduced with permission [105]. Copyright (2023) Elsevier. (**B**) Structure model of (**a**) CeO_2_, (**b**) CeO_2-X_, and (**c**) CN-CeO_2-x_. Ce, O, C, and N atoms as shown in green, red, pink, and purple. Charge energy difference of (**d**) CeO_2-x_ and (**e**,**f**) CN-CeO_2-x_. Reproduced with permission [106]. Copyright (2022) Elsevier.

**Figure 8 nanomaterials-13-02005-f008:**
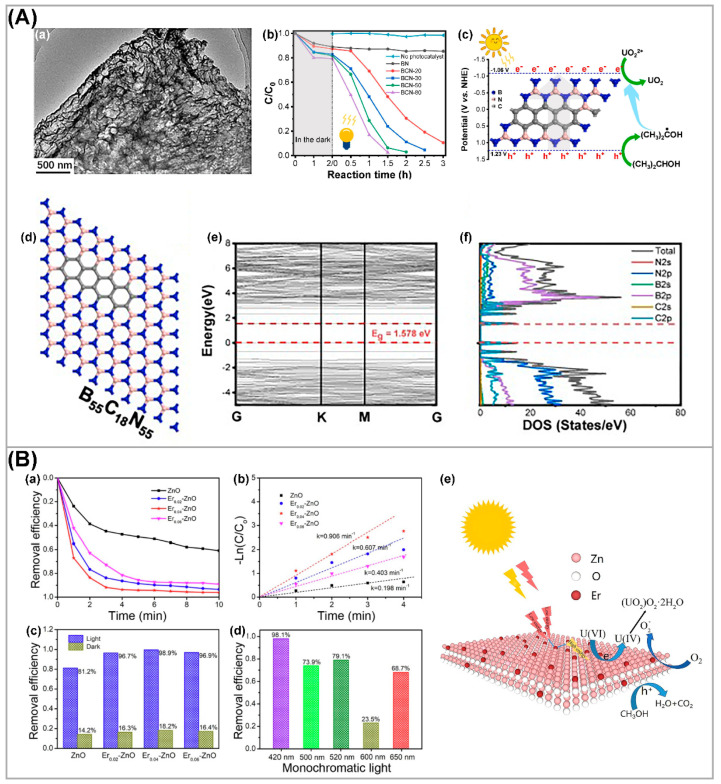
(**A**) (**a**) TEM image, (**b**) comparison of photoreduction capabilities, (**c**) and proposed mechanism for photocatalytic reduction in U(VI) over the BCN. (**d**) Optimized structures, (**e**) calculated band structures, and (**f**) corresponding density of states of BCN models. Reproduced with permission [108]. Copyright (2021) Elsevier. (**B**) (**a**–**d**) The photoreduction performance of U(VI) by ZnO, Er_0.02_-ZnO, Er_0.04_-ZnO, and Er_0.06_-ZnO, and (**e**) the proposed mechanism of U(VI) photoreduction by Er-ZnO. Reproduced with permission [110]. Copyright (2022) Elsevier.

**Figure 9 nanomaterials-13-02005-f009:**
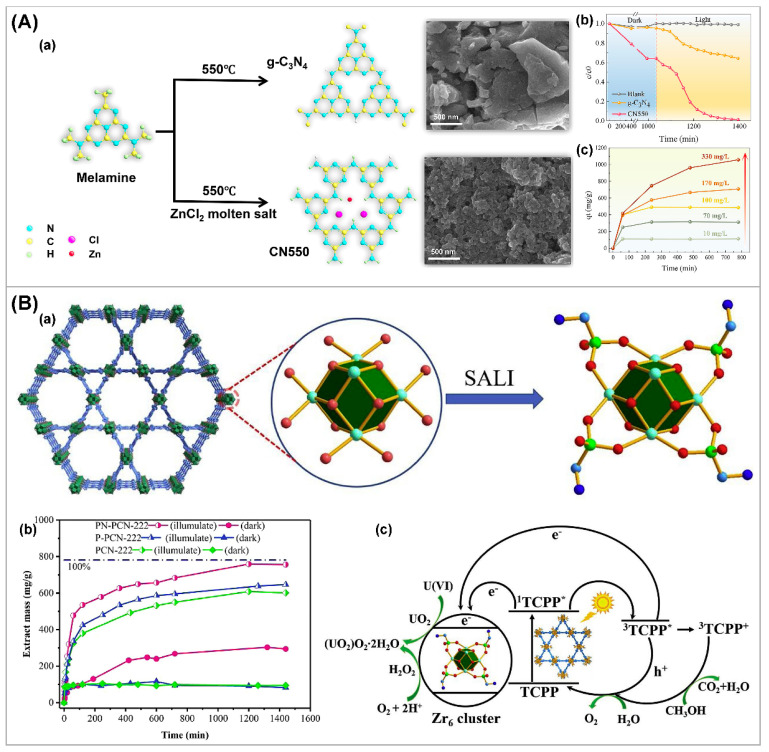
(**A**) (**a**–**c**) Scheme for the formation of g-C_3_N_4_ and CN550 with poly(triazine-imide) (PTI) structure and SEM images for both materials, (**b**) the variation in relative UO_2_^2+^ concentration vs. illumination time of 0.2 g L^−1^ CN 550 and 1 g L^−1^ g-C_3_N_4_ as photocatalysts, (**c**) uranium extraction from spiked seawater using SUPER method with initial uranium concentrations of 330, 170, 100, 70, and 10 mg L^−1^. Reproduced with permission [111]. Copyright (2021) Elsevier. (**B**) (**a**) Schematic representation of SALI (P: bottle green; C: Cambridge blue; N: blue; O: red; and Zr: green), (**b**) photocatalytic uranium extraction performance under different conditions, and (**c**) schematic illustration of selective enrichment and photocatalytic reduction in U(VI) based on PN-PCN-222. Reproduced with permission [27]. Copyright (2019) Elsevier.

**Figure 10 nanomaterials-13-02005-f010:**
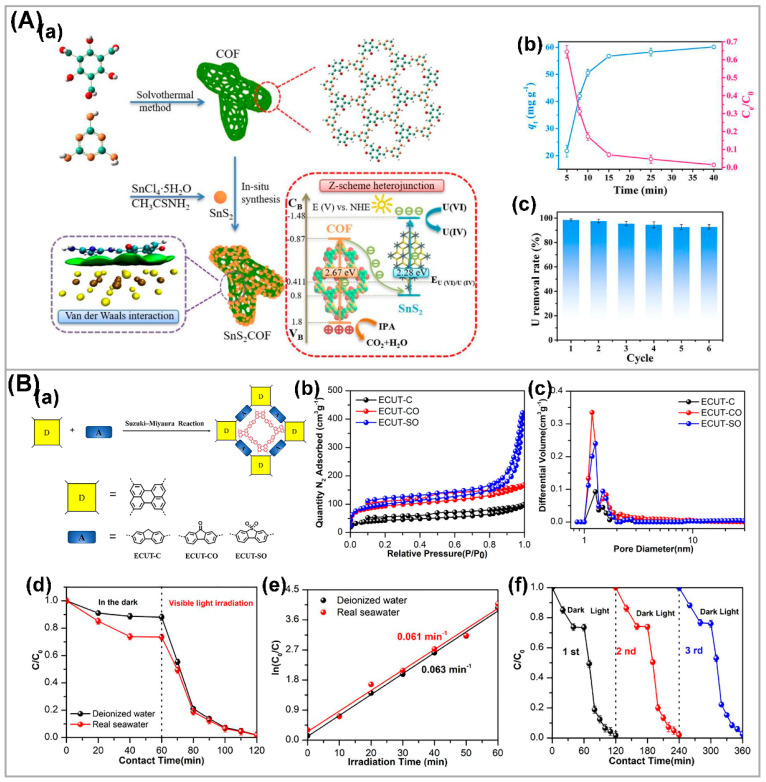
(**A**) (**a**) The mechanism diagram of SnS_2_COF used as a catalyst for photocatalytic reduction in U (VI), (**b**) photocatalytic removal capacity and removal rate of uranium from rare earth tailings wastewater by SnS_2_COF, and (**c**) cyclic performance of U (VI) photoreduction. Reproduced with permission [28]. Copyright (2023) Elsevier. (**B**) (**a**) Design principles and chemical structure of conjugated microporous polymers, (**b**) nitrogen adsorption–desorption curves, and (**c**) corresponding pore size distributions of CMPs; (**d**–**f**) photocatalytic reduction in UO_2_^2+^ over a ECUT-SO sample under visible-light irradiation. Reproduced with permission [6]. Copyright (2021) Elsevier.

**Figure 11 nanomaterials-13-02005-f011:**
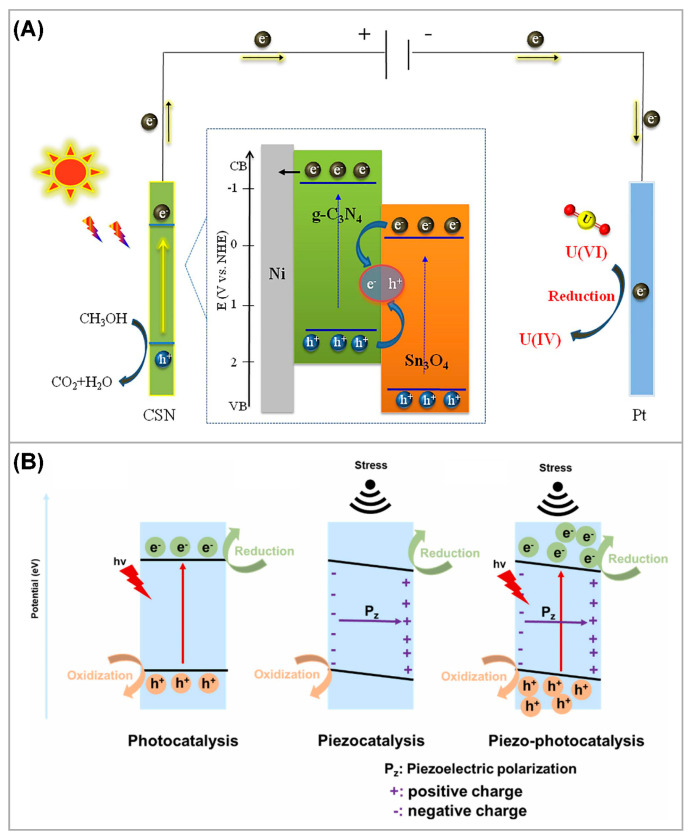
(**A**) Schematic illustration of the mechanism of photoelectrocatalytic reduction in U(VI) by CSN electrode. Reproduced with permission [131]. Copyright (2022) Elsevier. (**B**) Schematic diagram of the mechanism of photocatalysis, piezocatalysis and piezo-photocatalysis. Reproduced with permission [132]. Copyright (2023) Elsevier.

**Table 1 nanomaterials-13-02005-t001:** The photocatalytic U(VI) reduction performance over different catalysts.

Photocatalysts	C_U(VI)_ ^a^	pH	Light	RR (%) ^b^ t (min) ^c^	Ref.
TiO_2_/Fe_3_O_4_	0.1 mM	4.0	UV light (100 W high-pressure mercury lamp)	100, 30	[35]
GO/KTO	0.21 mM	6.0–8.0	UV-visible light (500 W Xe lamp)	100, 60	[36]
TiO_2_	0.2 mM	5.0	UV light (350 W mercury discharge lamp)	100, 100	[37]
C_3_N_5_/RGO	10 mg L^−1^	5.0	300 W Xe lamp (λ ≥ 425 nm, 3.08 mW/cm^2^)	94.9, 100	[38]
ZnO/rectorite	5 mg L^−1^	---	300 W Xe arc lamp	75, 150	[39]
SrTiO_3_/TiO_2_ electrospun nanofibers	100 mg L^−1^	4.0	400 nm with monochromatic light	-	[40]
SiO_2_/C nanocomposite	100 mg L^−1^	5.0	UV–vis absorption spectra	94.2, 120	[41]
S-g-C_3_N_4_	0.12 mM	7.0	visible light (A 350 W Xe lamp with a 420 nm cutoff filter)	95, 20	[42]
g-C_3_N_4_	27 mg L^−1^	6.0	300 W Xe lamp (λ ≥ 420 nm)	100, 20	[31]
C_3_N_4_	20 mg L^−1^	4.0–8.0	300 W Xe lamp (λ ≥ 420 nm)	92, 60	[43]
Nb/Ti NFs	20 mg L^−1^	5.0	simulated solar light (450 W xenon lamp)	90, 240	[44]
CdS/TiO_2_	50 mg L^−1^	6.0	solar simulator with a 420 nm cut-off filter	97, 240	[45]
g-C_3_N_4_/GO	80 mg L^−1^	5.0	300-W Xe lamp	-	[46]
Fe_3_O_4_@PDA@TiO_2_	50 mg L^−1^	8.2	---	73, 300	[47]
ZnFe_2_O_4_/g-C_3_N_4_	20 mg L^−1^	5.0	8 W LED light	94.62, 2400	[48]
MoS_2_/g-C_3_N_4_	50 mg L^−1^	4.5	photocatalytic reaction box (CEL-HXF 300, Beijing Aulight Co. Ltd.)	81.9, 120	[49]
graphene-like S-C_3_N_4_	0.12 mM	7.0	sodium hydroxide aqueous solution. A 350 W Xe lamp (λ ≥ 420 nm)	92, 30	[50]
Ti_3_C_2_/SrTiO_3_	0.21 mM	4.0	UV-visible light (wavelength, 320–2500 nm, 300 W Xe lamp)	80, 180	[51]
MoS_2_/THS	50 mg L^−1^	6.0	300 W Xe lamp equipped with an ultraviolet cutoff filter (λ ≥ 420 nm)	98, 80	[52]
bentonite/Portland cement composite	50 mg L^−1^	---	---	96.97, 1140	[53]
g-C_3_N_4_/TiO_2_	0.25 g·L^−1^	6.9	UV-visible light (wavelength, 320–780 nm, 300 W Xe lamp)	∼80, 240	[54]
CuO/CuFeO_2_	0.000126 mM	8.2	photoelectrochemical method: constant potential (−0.6 V vs. SCE) + simulated sunlight (150 W Xenon arc lamp with an AM1.5 G filter)	100, 90	[55]
TiO_2_/Fe_3_O_4_	13.5–108 mg L^−1^	3.5–9.3	100 W high—pressure mercury lamp (λ = 365 nm)	100, 30	[35]
PCN-222 MOF	500 mg L^−1^	3–11	300 W Xe lamp (λ ≥ 420 nm)	100, 5	[27]
Ti_3_C_2_/CdS	25–200 mg L^−1^	4–10	300 W Xe lamp (λ ≥ 420 nm, 500 mW/cm^2^)	97, 40	[56]
ECUT-SO	50 mg L^−1^	4	300 W Xe lamp (λ ≥ 400 nm)	97.8, 60	[6]
Sn/In_2_S_3_	16,200 mg L^−1^	3–9	500 W Xe lamp (λ ≥ 400 nm)	95, 40	[57]
PCB/g-C_3_N_4_	100 mg L^−1^	4	300 W Xe lamp (λ ≥ 400 nm)	---	[58]
TiO_2_	400 mg L^−1^	5.5	UV light (400 W mercury discharge lamp)	95, 135	[59]
B-g-C_3_N_4_	500 mg L^−1^	7.0	visible light (A 500 W Xe lamp with a 420 nm cutoff filter)	93, 20	[60]
isotype g-C_3_N_4_	0.168 mM	5.32	---	98, 20	[61]
BiOBr@TpPa-1	30 mg L^−1^	2.0–7.0	500 W Xe lamp (λ ≥ 420 nm)	91, 540	[62]
g-C_3_N_4_/LaFeO_3_	0.1 mM	5.0	300 W Xe lamp (AM 1.5 G)	94, 120	[63]
BC-MoS_2-x_	8 mg L^−1^	5.0	300-W Xe lamp with AM 1.5 G	92, 70	[33]
ZIF-8/g-C_3_N_4_	10 mg L^−1^	---	300 W Xe lamp (λ ≥ 420 nm)	100, 30	[64]
CdS/CN-33	0.1 mM	6.0	500 W Xe lamp (λ ≥ 420 nm)	100, 6	[26]
SnO_2_/CdCO_3_/CdS	50 mg L^−1^	---	500 W Xe lamp (λ ≥ 420 nm)	100, 70	[65]
Te/SnS_2_	8 mg L^−1^	4.8	300 W Xe lamp with AM 1.5 G filter	98.6, 90	[66]
Ag-doped SnS_2_@InVO_4_	60 mg L^−1^	6.0	450 W Xe lamp (λ ≥ 400 nm)	97.6, 100	[67]
MIL-53 (Fe)	0.21 mM	4.5	visible light (285 W Xe lamp, λ ≥ 420 nm)	80, 120	[68]
TiO_2_	0.21 mM	2.7	15 W black-light fluorescent bulb (λ = 320∼400 nm)	50, 780	[69]
CdS_0.95_Te_0.05_-EDA nanobelt	10–300 mg L^−1^	3–9	300 W Xe lamp (λ ≥ 420 nm)	97.4, 80	[70]
graphene aerogel	95.2 mg L^−1^	5.0	350 W Xe lamp (λ = 420– 800 nm, 160 mW/cm^2^)	96, 200	[71]
TiO_2_	0.42 mM	3.0	150 W Xenon arc lamp with an AM 1.5 G filter	80, 240	[72]

Note: ^a^ U(VI) concentration. ^b^ RR is removal rate and ^c^ t is time.

## Data Availability

Not applicable.

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
