# Peer review of "Advanced Photocatalytic Uranium Extraction Strategies: Progress, Challenges, and Prospects"

_nanomaterials, 2023, doi:10.3390/nano13132005_

Round 1

Reviewer 1 Report

In this current manuscript, the authors presented a systematic review of photocatalytic removal/extraction of Uranium species. They have addressed all the relevant points related to the topic for a better understanding of the topic and current scenario of this research. The review is nicely written and easy to understand. However, I was wondering about some headings that are just like the recently published one, please see Coordination Chemistry Reviews, 2022, 467, 214615. The authors need to read and compare their manuscript with this carefully and modify it accordingly. Also, please incorporate some more literature reports in Table 1 with some current references.

Reviewer 2 Report

This is an interesting and relevant manuscript. Please find comments to further improve this work below:

-line 33-36: why is that? In my understanding a nuclear power plant does not continuously release radioactive wastewater if it operates properly?

-line 43: uranium is not extremely rare, it is actually pretty abundant, hence the large quantities in seawater you mentioned earlier

-line 56: please provide a number for the low cost assumption presented here. How much is it in USD/lb?

-for me you need to better distinct if you want to recover uranium from unconventional resources or clean wastewater from uranium, or is it both? Maybe provide an overview about the range of uranium concentrations we are talking about. In seawater the uranium concentrations are only 3 micrograms per liter (I think), in some wastewater they can be much higher I think, and also in phosphoric acid in China there seem to be considerable concentrations (https://doi.org/10.1016/j.rser.2021.110740). In what concentration range can your technology be used? I think this would be very interesting for a review.

-Fig. 1 is great

-Maybe increase the size of the pictures in Fig. 5 or make this different Figures. Maybe that is just me because I am old, but it is really hard for me to read. Fig. 5D, I cannot read…

-probably in Chapter 5.4 I would really want a discussion about the economics of this. Is it really as cost effective as claimed in the abstract and introduction, and if so why is it not used yet?

This is research so I think there is no harm in acknowledging that it is not as economic as uranium mining yet, but I feel it should be discussed somewhere, especially in a review where we stand with this technology.

Generally great work, I learned something and hope my comments are helpful.

Round 2

Reviewer 2 Report

I don't see my comments addressed. Please go back and try to address the comments I gave you in the first revision.

I also see some logical flaws in the manuscript. In the introduction you claim that there are large emissions of uranium from nuclear power production, in the discussion on the economics of your process you say that the concentrations are actually to low to allow an economic application of the technology...

Both at the same time does not make sense to me.

The references you provided about the release of uranium from nuclear power plants considers nuclear accidents and not the standard operation of a nuclear power plant. It makes still sense to use your technology for wastewater treatment. It should be explained better then. At present it seems that you are expecting uranium release from nuclear power power plants during normal operation, which is not the case, and also not supported by the references you provided.

So the big question what your technology is suppose to do:

Wastewater treatment or uranium recovery?

has not been answered properly yet. There are tremendous efforts in China to recover uranium from unconventional resources such as phosphates (https://doi.org/10.1016/j.rser.2021.110740) if your work goes into this direction, good. So mention it. If you rather look at wastewater treatment, also good. But then please explain that better.

Please try to provide quantitative analysis about the expected uranium concentrations in the media you want to treat, and please provide a cost estimate (also quantitatively) about the uranium extraction costs. Simply saying that it is a low-cost technology, does not really help. Sorry...

Looking forward to receiving your comments, and seeing this published shortly.

Author Response

We have replied all comments from reviewer, please see the attached file.

Round 3

Reviewer 2 Report

Comments have been addressed. Thank you. This manuscript can be accepted for publication now.